

# Some methods to improve the utility of conditioned Latin hypercube sampling

Brendan P. Malone[1,2], Budiman Minansy[2] and Colby Brungard[3]

[1] CSIRO, Agriculture and Food, Canberra, ACT, Australia
[2] The Sydney Institute of Agriculture, The University of Sydney, Sydney, NSW, Australia
[3] Plant and Environmental Sciences, New Mexico State University, Las Cruces, NM, USA

## ABSTRACT

The conditioned Latin hypercube sampling (cLHS) algorithm is popularly used for planning field sampling surveys in order to understand the spatial behavior of natural phenomena such as soils. This technical note collates, summarizes, and extends existing solutions to problems that field scientists face when using cLHS. These problems include optimizing the sample size, re-locating sites when an original site is deemed inaccessible, and how to account for existing sample data, so that under-sampled areas can be prioritized for sampling. These solutions, which we also share as individual R scripts, will facilitate much wider application of what has been a very useful sampling algorithm for scientific investigation of soil spatial variation.

## INTRODUCTION

The conditioned Latin hypercube sampling (cLHS) algorithm (*Minasny & McBratney, 2006*) was designed with digital soil mapping (DSM) in mind. cLHS is a random stratified procedure that choses sampling locations based on prior information pertaining to a suite of environmental variables in a given area. cLHS has been used extensively in DSM projects throughout the world with recent examples in the last 5 years including *Sun et al. (2017)* in China, *Jeong et al. (2017)* in Korea, *Scarpone et al. (2016)* in Canada, and *Thomas et al. (2015)* in Australia. cLHS has also been used for other purposes and contexts too. For example, in optimal soil spectral model calibration (*Ramirez-Lopez et al., 2014*; *Kopačková et al., 2017*), understanding the conditions which determine Phytophthora distribution in rainforests (*Scarlett et al., 2015*), and assessing the uncertainty of digital elevation models derived from light detection and ranging technology (*Chu et al., 2014*).

For DSM, the algorithm exploits collections of environmental variables pertaining to soil forming factors and proxies thereof (*McBratney, Mendonça Santos & Minasny, 2003*; e.g., digital elevation model derivatives, remote sensing imagery of vegetation type and distribution, climatic data, and geological maps) to derive a sample configuration (of specified size), such that the empirical distribution function of each environmental variable is replicated (*Clifford et al., 2014*). Presuming that soil variation is a function of the chosen environmental variables, it is reasoned that models fitted using data collected

Corresponding author
Brendan P. Malone,
brendan.malone@sydney.edu.au

via cLHS, capture all the soil spatial variability and will be applicable across the whole spatial extent to be mapped.

However, our own experience, and from personal communication with other researchers and field technicians, a common set of methodological questions arise when using cLHS. These questions are:

1. How many samples should I collect?
2. Where else can I sample when a cLHS location cannot be visited because of difficult terrain, locked gate, safety reasons etc.?
3. How do I account for existing samples when designing a new survey?

The purpose of this technical note is to describe solutions to each of these questions. We first begin with a brief overview of the cLHS algorithm and then address each question separately.

## MATERIALS AND METHODS

### A short overview of cLHS

Conditioned Latin hypercube sampling is one of the many environmental surveying tools available for understanding the spatial characteristics of environmental phenomena. Extended discussions about soil sampling, surveying, and monitoring of natural resources in a broad context can be found in seminal publications such as *de Gruijter et al. (2006)* and *Webster & Lark (2013)*. cLHS has its origins in Latin hypercube sampling (LHS) first proposed by *McKay, Beckman & Conover (1979)*. LHS is an efficient way to reproduce an empirical distribution function, where the idea is to divide the empirical distribution function of a variable, $X$, into $n$ equi-probable, non-overlapping strata, and then draw one random value from each stratum. In a multi-dimensional setting, for $k$ variables, $X_1, X_2, \ldots, X_k$, the $n$ random values drawn for variable $X_1$ are combined randomly (or in some order to maintain its correlation) with the $n$ random values drawn for variable $X_2$, and so on until $n$ $k$-tuples are formed, that is, the Latin hypercube sample (*Clifford et al., 2014*). Its utility for soil sampling was noted by *Minasny & McBratney (2006)*, but they recognized that some generalization of LHS sampling was required so that selected samples actually existed in the real world. Subsequently, they proposed a conditioning of the LHS, which is achieved by drawing an initial Latin hypercube sample from the ancillary information, then using simulated annealing to permute the sample in such a way that an objective function is minimized. The objective function of *Minasny & McBratney (2006)* comprised three criteria:

1. Matching the sample with the empirical distribution functions of the continuous ancillary variables
2. Matching the sample with the empirical distribution functions of the categorical ancillary variables; and
3. Matching the sample with the correlation matrix of the continuous ancillary variables.

See *Minasny & McBratney (2006)* for full detailing of the cLHS algorithm.

## How many samples should I collect?

Ironically this question arose from a situation where we examined data from a ~100 ha field. Within this field 238 soil samples were collected by regular grid sampling. Soil samples were then analyzed for key soil properties. Also collected from the field were ancillary data including soil conductivity (from an EM38 sensor), elevation, and crop yield data. Our interest was in determining whether the 238 soil samples adequately covered the ancillary data space and if an alternative sampling configuration using cLHS could be used to determine a specific sample number that would achieve the same data coverage as the 238 sites.

We acknowledge that for most contexts, practitioners will not experience the same situation we have just described, but will only have a suite of ancillary data and will need to estimate an optimal sample number.

The solution to choosing an optimal number of samples is to compare the empirical distribution functions of incrementally larger sample sizes with those of the population. The resulting output will display a typical exponential growth (or decay) curve, depending on what comparative metric is used, that plateaus at some point with increasing sample size. This allows one to invoke a diminishing returns-like rule to derive an optimal sample number. This approach was exemplified in both *Ramirez-Lopez et al. (2014)* for selection of a number of observations to optimize the fitting of soil spectral models, and *Stumpf et al. (2016)* for evaluating an optimal sample size to calibrate a DSM model.

Comparing the empirical distribution functions of the samples and the population can be done in a number of ways. *Stumpf et al. (2016)* compared the overall variance of the ancillary data from an entire region, hereinafter referred to as the population, with that of the sample. The application of this approach for categorical data was not established, but for categorical data it is only necessary to match the relative proportions of the categories. Other metrics for continuous variables could involve comparison of the quantiles of the empirical distributions. This would require comparing an absolute deviation between the quantiles (which could be absolute difference or Euclidean distance) for each ancillary variable, then deriving a unified value by combining the distances (weighted averaging) from each variable. Alternatively, comparison could be made of the principal components of the ancillary variables (using the approach described by *Krzanowski (1979)*), which would alleviate the need to first compute distances for each variable and then combine into a single measure of distance.

Probably a more standard approach for comparing empirical distribution functions is the *Kullback & Leibler (1951)* divergence which is specifically designed for comparing distributions (*Clifford et al., 2014*). The Kullback–Leibler (KL) divergence (also called relative entropy) compares the relative proportions of samples within each histogram bin of the distributions. The KL divergence can be computed as:

$$\text{KL} = \sum_i O_i \left( \log_e O_i - \log_e E_i \right) \tag{1}$$

where $E_i$ is the distribution of an ancillary variable for the population ($i$ is a histogram bin), and $O_i$ is the distribution of the same ancillary variable from a sample.

The KL divergence decreases towards zero as the population and sample distributions converge (*Clifford et al., 2014*) and can be defined for both categorical and continuous ancillary data. Signal processing is probably the most common use case for KL divergence, such as the characterization of relative entropy in information systems where the related Shannon's information criterion (*Shannon & Weaver, 1949*) is widely applied. Although useful, the KL divergence is not incorporated into the *Minasny & McBratney (2006)* cLHS algorithm, where instead comparison is made using the quantity: $\sum_i |O_i - E_i|$. This metric could also be useful for optimization of a sample size too, but the difference between it and the KL divergence is that it penalizes all departures from the population values equally. With the KL divergence, the penalty of missing a sample from the tail of a distribution is larger than the penalty of missing a sample close to the mode.

In the following, we demonstrate the usage of KL divergence to help identify an optimal sample size in the ~100 ha field previously described. The cLHS algorithm was run sequentially with increasing sample sizes beginning at 10 and finishing at 500. A step size of 10 samples was used. Each sample size was repeated 10 times to assess the dispersion about the mean KL divergence estimate. KL divergence was estimated using a bin number of 25 for each ancillary variable and then aggregated for all variables by calculating the mean. Note that the justification for selecting a bin number of 25 was to replicate the settings as implemented *Clifford et al. (2014)*. Our own small investigations indicated this bin number to be adequate for our data, but would recommend for other studies and contexts to evaluate the KL divergence response to various bin numbers via iteration. Too few bins may not adequately capture the main features of a distribution, giving misleading low divergence values. While too many bins maybe overly sensitive to noise and provide high KL divergence values in otherwise near matching distributions.

In this work, The R package *clhs* (*Roudier, 2011*) was used for performing the cLHS. The inputs into the *clhs* function included a table form (R data.frame) of the ancillary data that was collected in the field of interest, that is, the proximal sensed data and yield data. We used the default 10,000 iterations of the annealing process to run the function. A schematic of the sample size optimization algorithm is given in Fig. 1.

## Where else can I sample when a cLHS location cannot be visited because of difficult terrain, locked gate, safety reasons, etc.?

One of the biggest criticisms levelled at cLHS is the rigidness around the sampling configuration. For example, both *Thomas et al. (2012)* and *Kidd et al. (2015)* remarked on the lack of guidance on what to do when a site cannot be accessed. They arrived at the same conclusion, that rather than continuing with cLHC sampling, the more flexible sampling approach of fuzzy stratified sampling be used. *Kidd et al. (2015)* demonstrated that this alternative sampling approach was comparable to cLHS in their work. Nevertheless, once a sampling location selected by the cLHC algorithm is unable to be visited for any number of reasons, a replacement site needs to be selected. The introduction of replacement sites can potentially degrade the objective of cLHS if the replacement sites are arbitrarily selected in nearby locations that are accessible.
**1. Identify target area and collate ancillary data.**

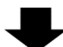

**2. Conditioned Latin hypercube sampling.**

- Repeat cLHC sampling *x* number of times, with sequentially increasing sample size. e.g. samples sizes for 10 to 500 in increments of 10

- For each sample size, record KL divergence [population ancillary data vs. sample ancillary data]

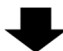

**3. Determine optimal sample size.**

- Make an xy-plot of number of samples vs. KL-divergence.

- Fit exponential decay function to plotted data.

- Derive cumulative density function (cdf) of the fitted exponential of 1 − KL divergence.

- Optimal sample size is the number of the cdf breaches the 95% probability level.

**Figure 1 Schematic of sample number size optimization algorithm.**

Modification of the cLHS objective function can be done so that accessibility is considered. This has been done by *Roudier, Beaudette & Hewitt (2012)* and *Mulder, De Bruin & Schaepman (2013)* where the cost of getting to sampling locations was incorporated into the annealing schedule of the cLHS algorithm. Such modifications, however, do not guarantee accessibility but do increase the probability that a site will be accessible. Adhering to an original cLHS site configuration, *Stumpf et al. (2016)* selected alternative sample sites by deriving numerous test sample sets based on the original cLHS sites with all possible combinations of the covariate data from the cLHS strata. They selected the test sample set that most closely represented the original cLHS sites via quantile matching. Implementing this method would require prior knowledge of accessible and inaccessible sites, which for *Stumpf et al. (2016)* was based on terrain slope and selected land use classes.

Neither of the approaches described above accommodate situations where issues of accessibility are found in the field, that is, site inaccessibility is unplanned.

*Clifford et al. (2014)* grappled with this issue and proposed a solution that involves simulated annealing for optimally selecting accessible sites from a region. This method involves multiple criteria such as the KL divergence, ease of access, and geographical coverage, which are all combined into a single criterion. While useful, this approach is computationally demanding to the extent that it would be difficult to apply directly in a field setting. To obviate the need for this, an ordered list of alternative sites close to each of the primary target sites (should the primary target site prove inaccessible) is produced prior to entering the field.

An easier, pragmatic option as demonstrated in the USA by *Brungard & Johnanson (2015)* would be to calculate a similarity measure to an inaccessible cLHS site within a given buffer zone. In their example, a Gower's dissimilarity index (*Gower, 1971*) was used. With this approach, areas with high similarity to the inaccessible cLHS site could then be identified and the new sample location moved to these areas if needed. Such an approach would allow alternative sites to be determined when in the field, given an appropriate computation device or if areas with high similarity to each location were generated before field sampling begins. This approach is also conveniently provided as a function in the clhs R package by *Roudier (2011)*.

In the following, we demonstrate a similar approach to that proposed by *Brungard & Johnanson (2015)*. It differs by generalization of the metric used to calculate distance between individuals for each of the variables, and it uses a membership function to derive estimates of similarity to a site that has been deemed inaccessible. Flexibility in the selection of distance metric allows alternatives other than the Gower distance to be considered.

We demonstrate this approach using a collection of 341 soil samples that cover the Hunter Wine Country Private Irrigation District (HWCPID), a 220 km$^2$ region in the Lower Hunter Valley Region, of New South Wales, approximately 140 km north of Sydney, Australia. The description of the sampling and data collection are described in *Malone et al. (2014)*. For this example, we used the following ancillary variables derived from a 25 m digital elevation model: elevation, slope, terrain wetness index, multi-resolution valley bottom flatness, and potential incoming solar radiation. We also used categorical rasterized data from a 1:100,000 geological unit map (*Hawley, Glen & Baker, 1995*), and a 1:250,000 legacy soil map depicting the surveyed soil units (*Kovac & Lawrie, 1990*) that cover the HWCPID. In this example, each of the 341 sampling locations were assumed to be inaccessible and alternative sites were selected for each.

We then compared the KL divergence between the original and alternative sample sites and assess whether the alternative sample sites capture the same environmental information using the following algorithm (also illustrated in Fig. 2).

Given an inaccessible site location:

1. Create a sampling zone (i.e., buffer) from which an alternative site can be selected and extract all the ancillary data from inside this zone. We used a circled sampling zone of radius 500 m.

2. Calculate the multivariate distance between the inaccessible site and all ancillary data in the sampling zone. If ancillary data is categorical, remove all areas of the sampling

zone that do not match the category at the initial sampling location. If ancillary data is continuous calculate a Mahalanobis distance.

3. Transform Mahalanobis distance to a similarity score using a negative sigmoid function:

$$\text{Similarity} = 1 - \frac{1}{1 + e^{(1 - \times (\text{dist}_i - \text{dist}_{\text{med}}))}} \quad (2)$$

Where dist is the Mahalanobis distance of the ancillary data $i$ and the inaccessible site and $\text{dist}_{\text{med}}$ is the median Mahalanobis distance of the available data within the buffer area. We used the median because it is less susceptible to outliers than the mean. The similarity is expressed on a range between 1 and 0 with numbers approaching 1 being highly similar, and is akin to a membership function.

4. Select an alternative site where the similarity is above a given threshold. In our example this threshold was set to 0.975. Alternatively, one could rank the sites (smallest to highest distance) and select a given number from the top for possible consideration. Whichever the approach, an alternative site may be selected at random from the sites that pass the threshold similarity value, or alternatively the nearest one to the inaccessible site could be selected and so on until it is found that access to the sampling location is possible. In our example, we selected the former option.

5. Repeat steps 1 to 4 for every inaccessible site location.

## How do I account for existing samples when designing a new survey?

This is the situation where existing soil samples exist within a sampling domain, and the user wants to derive a sampling configuration that captures environmental variation that the existing samples fails to achieve appropriately. With its current functionality, the clhs R package by *Roudier (2011)* goes some way to addressing this problem through forming $n1+n2$ marginal strata per covariate. Here $n1$ represents existing site data, and $n2$ represents marginal strata where new sampling locations can be derived. Essentially one selects a sample size of $n2$, adds in the existing sample data ($n1$) that have previously been collected from the study area, and then runs the clhs algorithm. This appears a useful solution but probably needs to be tested or compared with other approaches such as described below.

Possibly a more explicit approach to dealing with the aforementioned problem, is to first assess the environmental coverage of the existing samples, then look for gaps (where there is no coverage in the data space), and then prioritizing new samples to those areas as they occur in the field. This general approach was performed by *Stumpf et al. (2016)*. A more thoroughly described approach is the hypercube evaluation of a legacy sample (HELS) algorithm and its companion the HISQ algorithm described in *Carré, McBratney & Minasny (2007)*. The HELS algorithm is used to check the occupancy of the existing samples in the hypercube of the quantiles of given ancillary data, and to determine whether they occupy the hypercube uniformly or if there is over- or under-observation in partitions of the hypercube. The companion HISQ algorithm is used to preferentially select additional samples in areas where the degree of under-observation is

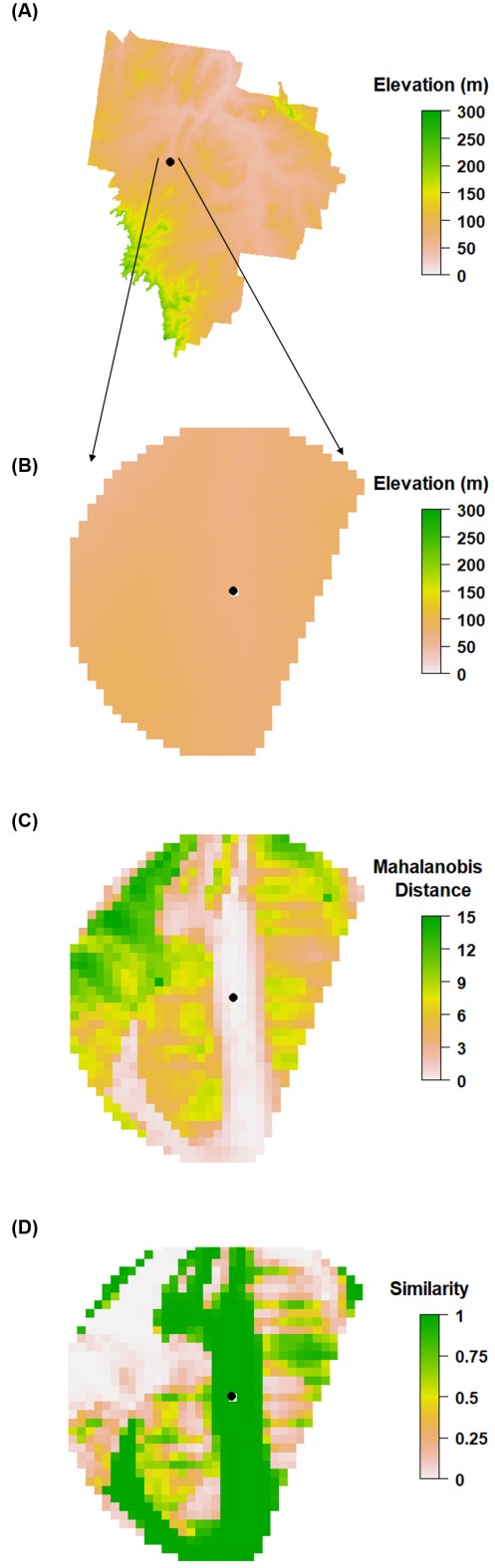

**Figure 2 Selection of alternate sites.** Illustrated process of the algorithm for selecting an alternative sampling site when a cLHS site is inaccessible. (A) A sample site (as indicated by a marked point) within the HWCPID area that has been determined to be inaccessible. (B) A circular buffer area is created around the site. If categorical data are being used, those categories that do not match that at the sampling

**Figure 2** (continued)
site are excluded. In the given example a circular buffer zone is used, but a portion was excluded due to non-matching of the categorical variables. (C) The Mahalanobis distance is estimated between the inaccessible site and all cells in the buffer area. (D) Similarity is estimated using Eq. (2) and ranges between 0 and 1. An alternative sample site is selected if the similarity exceeds 0.975.

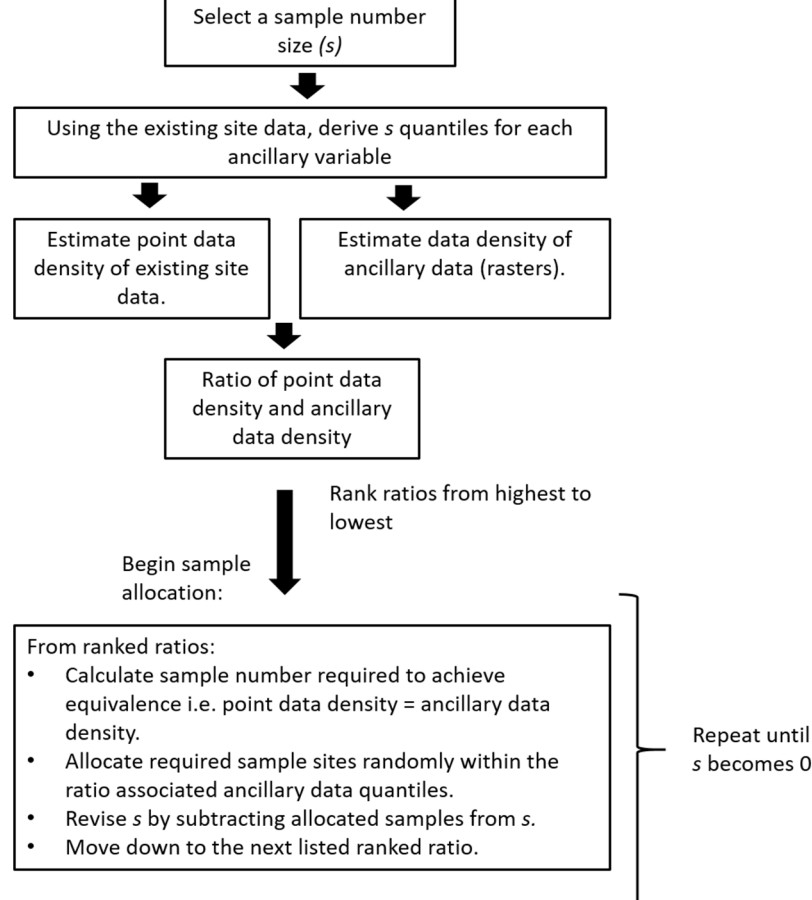

**Figure 3 Adapted HELS algorithm.** Schematic of the adapted HELS (aHELS) algorithm. To use this algorithm, one needs their collection of existing site data from the study area, together with a suite of ancillary data rasters. Finally, a specified sample size number (*s*) needs to be given.

greatest. A limitation of the HELS algorithm is that the full hypercube of all ancillary variables needs to be computed (i.e., if there are *k* number of variables, *k*-cubes need to be formed). We present an algorithm that is an adaptation and simplification of the combined HELS and HISQ algorithms.

This example uses the same ancillary data and the 341 sample sites from the HWCPID study area described earlier. For this example, we want to add an additional 100 samples while accounting for the prior 341 sample sites. We call this algorithm, adapted HELS (aHELS). A schematic of the aHELS algorithm is shown in Fig. 3 and described in the following:

0a. Select a sample size of size $s$. In this example $s = 100$

0b. Extract the ancillary data values for existing observations ($o$). In the example $o = 341$.

1a. Construct a quantile matrix of the ancillary data. If there are $k$ ancillary variables, the quantile matrix will be of $(s + 1) \times k$ dimensions. The rows of the matrix are the quantiles of each ancillary variable.

1b. Calculate the data density of the ancillary data. For each element of the quantile matrix, tally the number of pixels within the bounds of the specified quantile for that matrix element. This number is divided by $r$, where $r$ is the number of pixels of an ancillary variable.

1c. Calculate the data density of ancillary information from the existing legacy samples. This is the same as 1b except the number of existing observations are tallied within each quantile (quantile matrix from 1a) and the density is calculated by dividing the tallied number by $o$ instead of $r$.

2. Evaluate the ratio of densities. This is calculated as the point data density divided by the grid density.

3. Rank the density ratios from smallest to largest. Across all elements of the $(s+1) \times k$ matrix of the ratios from step 2, rank them from smallest to largest. The smallest ratios indicate those quantiles that are under-sampled, and should be prioritized for selecting additional sample locations. In this ranking, it is important to save the element row and column indexes.

4a. Begin selection of additional sample locations. Start by initiating a sample of size $s$

4b. While $s > 0$ and working down the ranked list of ratios:

5. Estimate how many samples ($m$) are required to make grid density = data density. This is calculated as $o \times$ the grid data density in the same row and column position as the density ratio.

6. Using the same row and column position of the quantile matrix from 1a, select from the grid data all possible locations that meet the criteria of the quantile at that position in the matrix. Select at random from this list of possible locations, $m$ sampling locations.

7. Set $s = s - m$, then go back to 4b.

The algorithm will terminate once $s$ sample locations have been selected. A caveat to using the aHELS algorithm is that it is not immediately amenable to optimizing sample size as described in the earlier section. Essentially, one needs to select a sample size, then run the algorithm.

Possibly a much more visual way—yet also complementary to the aHELS algorithm—to assess relatively adequate and under-sampled areas is to create a map that could show such patterns. We developed an algorithm called count of observations (COOBS) that can achieve this objective. The COOBS algorithm is implemented on a pixel basis and requires a stack of ancillary data pertaining to the environmental variables for a given spatial domain, and the associated legacy data points from the same spatial domain. The legacy points will have been intersected with ancillary data. We demonstrate this COOBS algorithm using the HWCPID example. A schematic of the COOBS algorithm is presented on Fig. 4.

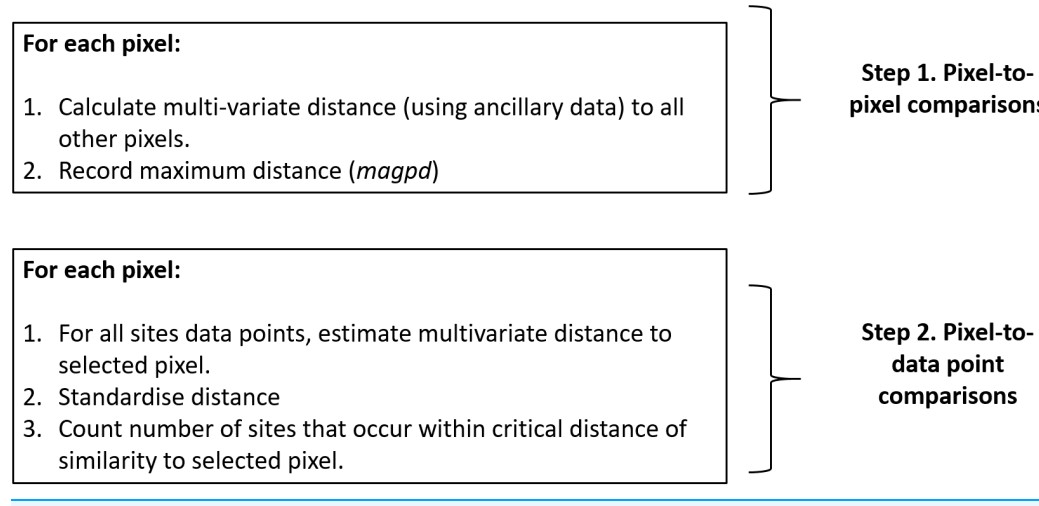

**Figure 4 COOBS algorithm.** Schematic of the COOBS algorithm. To use this algorithm, one needs their collection of existing site data from the study area, together with a suite of ancillary data rasters.

1. For each pixel location i:

1a. Calculate the multivariate distance to every other location in the stacked ancillary data. We use the Mahalanobis distance because it preserves the correlation between variables, but other distance or similarity metrics can be considered.

1b. Record the maximum distance, which we define as *magpd*. Note that the minimum distance will always be zero.

2. For each pixel location $i$:

2a. Calculate the multivariate distance of the ancillary data between the pixel $i$ and each of the legacy data points. We call this the data distance (*dd*).

2b. Calculate a standardized distance (*sdd*) for each legacy data point which is simply:

$$sdd_i = 1 - \frac{dd}{magpd_i}$$

As the *sdd* will scale from 0 to 1, values close to 1 means at least one observation point is similar to the ancillary data for the selected pixel.

3. Set a similarity threshold close to 1, then count the number of legacy observations that are equal to or better than that criteria. Save this number and pixel location together, then move to the next pixel, that is, go back to Step 2a.

From the above described COOBS algorithm, and depending on how it is implemented, the first step will create a raster of the target area depicting the spatial pattern of *magpd*. This map is then interrogated pixel-by-pixel in step 2 to estimate the associated COOBS number. Alternatively, the two steps of the COOBS algorithm could be implemented as a single workflow whereby the maps of *magpd* and COOBS are derived at near the same time as each other, pixel-by-pixel.

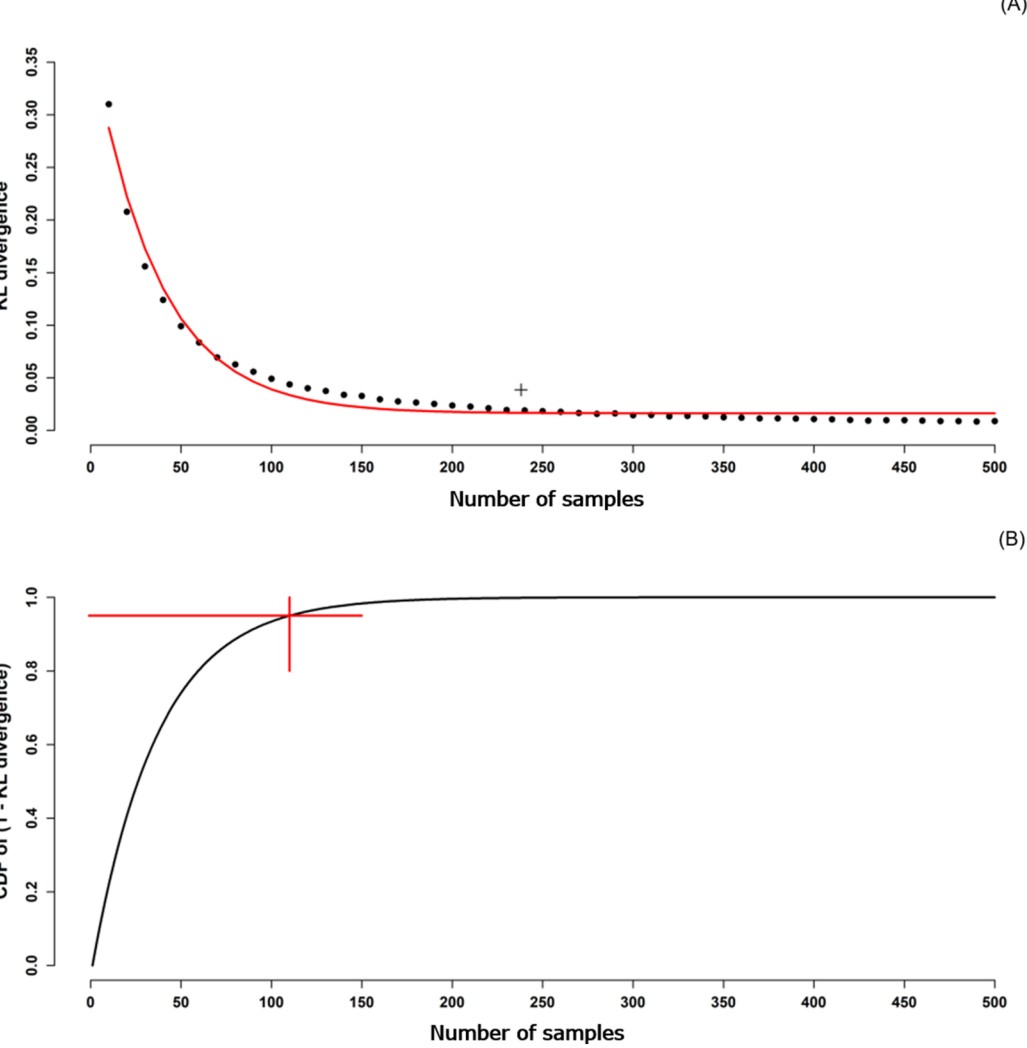

**Figure 5 KL divergence as function of sample size.** (A) KL divergence between ancillary data of a sample and ancillary data of the population as a function of sample size. The fitted line is a negative exponential decay function fit to the KL divergence with the relationship $y = b\_1 \cdot [\![\exp]\!] \,^{\wedge}((-kx))+b\_0$ where $x$ and $y$ were sample size and KL divergence respectively. The fitted parameters of $b\_0, b\_1,$ and $k$ were 0.275, 1.185, and 0.028, respectively. The "+" symbol is the KL divergence for a grid sample of 239. (B) Cumulative density function of the $1 - $ KL divergence (from A) as a function of sample size. The point of intersection of the two straight lines is the optimal sample size as this is the point where the cdf breaches 95%.

## RESULTS AND DISCUSSION

### How many samples should I collect?

The resulting output is shown in Fig. 5A which is a plot of the mean KL divergence with increasing sample size and is a classical exponential decay with increasing sample size. We do not show the standard deviation bars on this plot as they are very small with increasing sample size and are not visible on the plot. However, relatively larger standard deviations were observed at smaller sample sizes which decreased with increasing sample sizes. We also calculated the KL divergence for the 238 samples already

**Table 1 KL-divergence comparison between original sample and optimized sample.**

| Ancillary variable | KL-divergence (original) | KL-divergence (relocated) |
|---|---|---|
| Terrain wetness index (unitless) | 0.021 | 0.019 |
| Slope (°) | 0.038 | 0.041 |
| Multi-resolution valley bottom flatness (unitless) | 0.009 | 0.019 |
| Potential incoming solar radiation (kWh/m$^2$) | 0.063 | 0.052 |
| Elevation (m) | 0.034 | 0.033 |
| Mean | 0.033 | 0.033 |

**Note:**
KL-divergence of the original sample configuration and the relocated sample configuration (341 sites) for each of the ancillary data and an overall mean.

collected from the field of interest. This was found to be 0.039, and as can be seen on Fig. 5A as the "+" symbol. This mark is above the fitted line for the same sample size using cLHS. This indicates that (for this example), cLHS is superior to grid sampling for capturing the variation in ancillary data for the same sample size.

To estimate an optimal sample size, we derived a cumulative density function for the fitted line on Fig. 5A, and identified the sample size required to capture 95% of the cumulative probability (Fig. 5B). The optimal sample size was 110. The KL divergence for this sample size was 0.033, slightly lower (better) than that achieved with the actual 238 grid samples.

In a practical setting, the 110 samples would be used as the minimum recommended sample size. However, one caveat is that other metrics of comparison are likely to result in differing optimal sample sizes because the underlying calculation is different. In situations where multiple comparison metrics are used, it might be pragmatic to assess what the minimum and maximum optimal sample sizes are, then select a recommended sample size in the middle. Though likely not optimal, this approach would be an improvement on the arbitrary selection of sample size that currently occurs in most studies.

### Where else can I sample when a cLHS location cannot be visited because of difficult terrain, locked gate, safety reasons, etc.?

Table 1 presents the KL divergences for each ancillary variable for both the original sample configuration and the alternative sample configuration for each of the 341 sites. Overall, the mean KL divergence is equal for both sampling configurations which is a desired outcome of the algorithm. With this relatively simple procedure or adopting the similar approach described in *Brungard & Johanson (2015)* it becomes possible to derive alternative sample locations within the field at the time of sampling. This delivers to the field technician an objective way to select an alternative sample site (or possibly even a general area) during the survey campaign.

### How do I account for existing samples when designing a new survey?

Figure 6A shows a map with the original 341 sites and the additional 100 sites selected using the aHELS algorithm. The COOBS algorithm created the map on Fig. 6B.
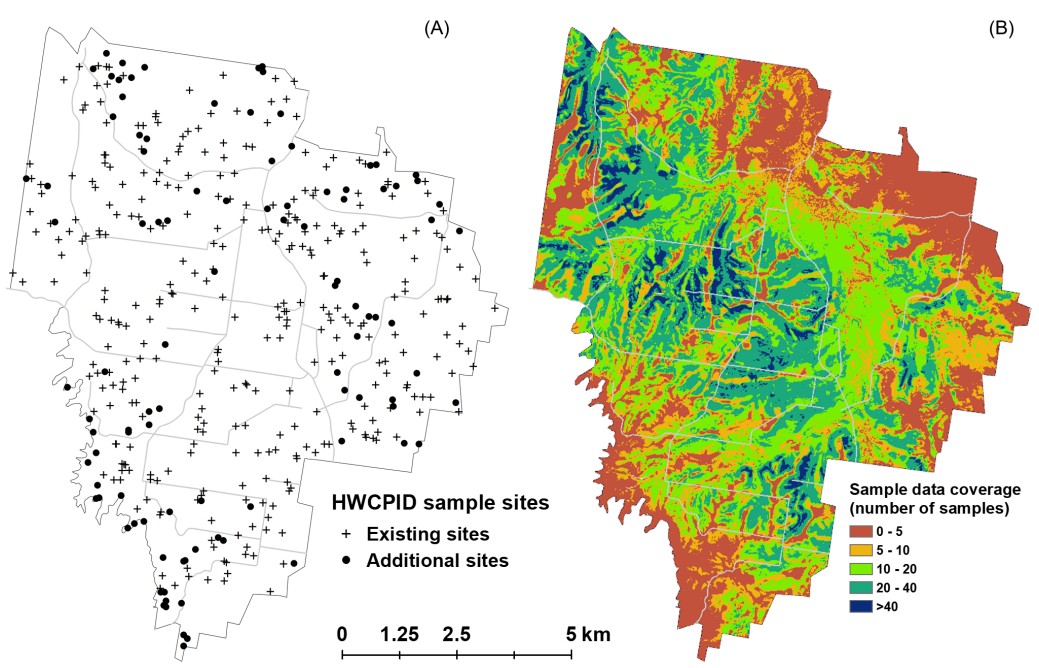

**Figure 6 Comparative outputs of aHELS and COOBS algorithms.** (A) HWCPID existing 341 sites (crosses) and additional 100 sites (dots) selected using the adapted HELS algorithm. (B) Existing sample data coverage using the 341 HWCPID sites based on using the COOBS algorithm. Values indicate the number of existing soil samples that are above a similarity threshold for each pixel. Lower values indicate areas with fewer similar observations and which likely require additional sampling.

A caveat to the COOBS algorithm though is that it is computationally heavy. Some of this burden can easily be diffused via compute parallelization, however. Alternatively, and in order to take advantage of spatial correlation, it could be possible to take a representative sample from the provided ancillary data of a specified size, calculate *magpd* for each of these locations, then interpolate this target variable across the whole area. Future investigations would need to assess the viability and efficiency of this approach.

Overall, both the aHELS and *COOBS* algorithms allow one to understand which areas in a spatial domain are adequately and under-sampled. The aHELS algorithm is explicit in selecting sites preferentially to locations in the environment that are not captured by the existing site data. Relatedly, the COOBS algorithm provides a visualization of general areas where under sampling is prevalent. As an aside, it may be possible to use the COOBS derived map to design an additional survey by constraining the cLHS algorithm to areas where the COOBS value is below some specified threshold. Nevertheless, both aHELS and COOBS are somewhat comparable in their objectives and ultimately generate similar outcomes as shown in Table 2 where the proportion of sample sites selected using the aHELS is much higher in the areas where the COOBS algorithm determined relatively low existing data coverage. For example, 87% of additional sites appear to be allocated in areas where the existing data coverage is ≤10 sites.

**Table 2 Similarity between adapted HELS and COOBS algorithm outputs.**

| Legacy sample data coverage (derived from *COOBS* algorithm) | Proportions of allocated adapted HELS sites (100) | Proportions of existing sites (341) |
|---|---|---|
| 0–5 | 0.66 | 0.26 |
| 5–10 | 0.21 | 0.17 |
| 10–20 | 0.12 | 0.30 |
| 20–40 | 0.01 | 0.23 |
| >40 | 0.00 | 0.03 |

Note:
Proportions of original and additional sampling sites in the HWCPID selected using the adapted HELS algorithm that occur within groupings of the sample data coverage, determined using the *COOBS* algorithm. We note that *COOBS* means at the pixel level, the count of observations estimated to be similar in terms of the given ancillary data.

## CONCLUSIONS

This technical note provides some solutions to common questions that arise when the cLHS algorithm is used for designing a soil or any other environmental survey. We have collated solutions that others have come up with to deal with such questions, and also presented new and/or modified solutions too. We have presented a solution to optimize a sample size number and to take into consideration existing soil samples. Importantly we have introduced a relatively simple approach to relocate a site for soil sampling in situations of inaccessibility without deteriorating the original cLHS site configuration. R scripts with associated data examples for each of these solutions is shared in a data repository at: https://bitbucket.org/brendo1001/clhc_sampling.

### Funding
The authors received no funding for this work.

### Competing Interests
Budiman Minansy is an Academic Editor for PeerJ.

### Author Contributions
- Brendan P. Malone conceived and designed the experiments, performed the experiments, analyzed the data, contributed reagents/materials/analysis tools, prepared figures and/or tables, authored or reviewed drafts of the paper, approved the final draft.
- Budiman Minansy conceived and designed the experiments, analyzed the data, contributed reagents/materials/analysis tools, authored or reviewed drafts of the paper, approved the final draft.
- Colby Brungard authored or reviewed drafts of the paper, approved the final draft.

### Data Availability
Bitbucket: https://bitbucket.org/brendo1001/clhc_sampling/src/master/.

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
