# Peer review of "Some methods to improve the utility of conditioned Latin hypercube sampling"

_PeerJ, doi:10.7717/peerj.6451_

## Round 0.1 · original submission · Minor Revisions

· Academic Editor

Minor Revisions

There are some minor issues that could be addressed for improving the readability of the paper.

They mainly concern changes to the figures (e.g. for showing the additional sample data coverage across the study area in fig.3) and in table 2 (where it could be useful to see some additional information about the ancillary data used).

Moreover, it is also suggested to include three separate flowcharts for algorithms 1–3.

Reviewer #3 suggests that a worked example with real data near a cLHS-proposed sampling site would be a nice addition.

Reviewer 1 ·

Basic reporting

It is clear.

Experimental design

It is clear.

Validity of the findings

It is clear.

Reviewer 2 ·

Basic reporting

The English of the manuscript is clear to the readers and sufficient references are provided.
The article structure meets the standard and R scripts of the study are provided.
The results confirm the aims/hypotheses of the paper.

Experimental design

The article is novel because it provides modifications to a previous widely used sampling algorithm proposed by one of the authors.

The research questions are well defined and the three problems related to sampling identified in the article are also widely faced in other field studies.

The use of various statistical indices are rigorous and provides useful insights into the validity of the new algorithms the author proposed.

The R scripts provided by the authors enable others to repeat the results.

Validity of the findings

The data, methods and results and robust for the first two algorithms and case studies.

Additional statistical metrics should be provided for the third case study.

Additional comments

In this study, the authors collated, summarized, and extended existing solutions to problems that field scientists face when using cLHS sampling. They provided solutions to optimize the sample size, re-locate sites when an original site is inaccessible, and to account for existing sample data when additional samples are to be taken. The authors also provided the R scripts to facilitate other researchers with the sampling design in the field.

In general, the manuscript was well written and the ideas are novel to the audience in the related fields. I believe that the manuscript will interest a large number of researchers who face similar problems in using the popular cLHS sampling in the field and provide guidance for people who attempt to improve the existing random stratified sampling algorithms.

I have a few minor comments on the manuscript.

Lines 134-135: Why the penalty of missing a sampling from the tail of a distribution is larger than that close to the mode? When the sample has a small distribution, O_i should be smaller, isn’t it?

Figure 1: the legends are too small to read.

Figure 3: Please provide a new subfigure that shows the additional sample data coverage across the study area.

Table 2: Please provide the proportions of the existing sample data coverage. Please also provide the mean values of the ancillary data calculated from the existing samples, additional samples, and total samples as well as the mean values of the ancillary data across the whole study area.

It will be better for the authors to include three separate flowcharts for algorithms 1–3. This way, the readers can understand the algorithms more easily.

Reviewer 3 ·

Basic reporting

The basic pieces of this manuscript are in place and flow together nicely. There are some grammatical errors (e.g. lines 377-378) that need to be fixed. Another example is on line 403: soil samples are not being re-located, the sampling location is. I recommend one more proof-reading to find these errors. They are small but jarring.

The background on KL divergence was helpful. I suggesting linking it to the concept of entropy if possible in 1-2 sentences and suitable citation. Reviewing the literature on entropy helped me understand how the authors are using KL divergence.

I think that it is important to elaborate on the fact that KL divergence is based on probability distributions--which have to be estimated by histogram / binning when applied to continuous variables. Number of bins, bin size, and interval selection are critical aspects of this step and need to be described. A couple of sentences with a concrete example would likely bring more readers closer to the methods.

It is not clear to me if the statement on line 141 is correct: "...bin size of 25 (effectively 25 quantiles)...". 25 equally-spaces bins dividing a distribution are not the same as the locations of 25 quantiles estimated from said distribution. Is there an assumption that has been left out of the description whereby the bin width is not equal? Please elaborate.

Lines 192--194 are confusing: Gower's (generalized) distance metric applies to collections of nominal, ordinal, and ratio data. What are the differences between "numerical and continuous variables" (line 194)?


The topic of sampling and sample location selection has been heavily influenced by the work of de Gruijter, J., Brus, D.J., Bierkens, M.F.P., Knotters, M. -- consider mentioning some of this work and citing either their book or specific papers.


Tables need to have units listed for each column: Table 2 is very hard to understand due to this omission.

Figure 1 would benefit greatly from labels near color legends. What do they mean and what are the units (if appropriate). Figures 1 [a,b,c] could benefit from a line denoting the 500m buffer. It was not clear at first as to what the non-circular shape represented.

Figure 2, consider changing "sample number" to "Number of Samples".

Experimental design

Commenting on specific methods here since there is no experimental design.

Lines 216-222: Why not just use distance? Converting to a similarity score seems like further abstraction of an otherwise abstract notion. Please elaborate.

Line: 224 What is the significance of the selected threshold? Why is a threshold needed when a ranked list of alternatives is the goal?

Lines 309-322:

This is an interesting idea although without tremendous computing power not likely the be of practical use to the intended audience. Imagine a situation where the scientist has 6 10,000 x 10,000 pixel maps of interest--that is a lot work for a single (?) maximum distance. Given the spatial correlation in most environmental data, 1/10th -- 1/100th of the original pixels could probably be used to compute "magpd".

Step 2 in this section, is this computed per pixel or over all pixels?

I expect dd to have a very large range and deviate from a normal distribution. Would the equation on line 317 be overly biased by the distribution of distances?

Validity of the findings

Overall this is a nice synthesis of what has been done, what is possible with current software (cLHS package for R), existing shortcomings, and some reasonable improvements.

The paper is written as a technical note but provides very little in terms of concrete example and viable implementation.

A worked example with real data near a cLHS-proposed sampling site would be a nice addition. Ultimately someone using cLHS and related methods should have a practical understanding of the results. Referencing the Brungard and Johnanson paper is one avenue but an additional figure and commentary would be more useful. This example would also provide an opportunity to inject some reality into the discussion: distances a fine way to rank pixels but fail to accommodate our understanding of the world. We do not go out into the wild blindly searching for sampling locations.

As for implementation: would the authors consider submitting material to the cited cLHS package for R? This would be a fine way to make the theoretical available to the target audience. Typically, methods papers cite an implementation that is already been made available or is available post-publication.

Additional comments

I think that the methods outlined in this paper will be of great interest to those in the fields of Earth and environmental sciences. The popularity of the cLHS algorithm is evidence enough.

In addition to my comments above I would like to add one more thing, more of a philosophical comment. Anyone who has worked in the field understands that pre-specified sampling locations are more useful as suggestions. Access, local (unexpected) disturbances, etc. all contribute to re-evaluation of what otherwise seemed like a great place to dig a hole. Furthermore, we never have a complete "stack" of covariates that describe the processes we are trying to wrangle. Therefore, it might be helpful to include a a reminder that cLHS and related sample-optimization strategies leave room for _more_ thought about how limited resources should be spent, and should not replace subject expertise or experience when theory meets practice. I am sure you all can make this point more effectively than I just did here.

---

## Round 0.2 · accepted · Accept

· Academic Editor

Accept

In the revised version of the manuscript very detailed responses are provided to comments raised by all the reviewers. Requested changes have been mainly applied and most important choices have been fully discussed both in the rebuttal letter and in the revised text (see e.g. the question raised by the reviewer about the structure of Table 2). I think that the ms is now suitable for publication in PeerJ.

Reviewer 2 ·

Basic reporting

The English of the manuscript is clear to the readers and sufficient references are provided.
The article structure meets the standard and R scripts of the study are provided.
The results confirm the aims/hypotheses of the paper.

Experimental design

The article is novel as it provides modifications to a previous widely used sampling algorithm proposed by one of the authors.
The research questions are well defined and the three problems related to sampling identified in the article are also widely faced in other field studies.
The use of various statistical indices are rigorous and provides useful insights into the validity of the new algorithms the author proposed.
The R scripts provided by the authors enable others to repeat the results.

Validity of the findings

The data, methods, and results are robust for the algorithms and case studies.

Additional comments

The authors have provided detailed responses to comments raised by all the reviewers. The manuscript has been materially improved and become much clearer to the readers after extra figures and tables have been inserted.